# Conformational specificity of opioid receptors is determined by subcellular location irrespective of agonist

**Stephanie E Crilly[1,2], Wooree Ko[2†], Zara Y Weinberg[2‡], Manojkumar A Puthenveedu[1,2]\***

[1]Cellular and Molecular Biology Program, University of Michigan, Ann Arbor, United States; [2]Department of Pharmacology University of Michigan Medical School, Ann Arbor, United States

**Abstract** The prevailing model for the variety in drug responses is that different drugs stabilize distinct active states of their G protein-coupled receptor (GPCR) targets, allowing coupling to different effectors. However, whether the same ligand generates different GPCR active states based on the immediate environment of receptors is not known. Here we address this question using spatially resolved imaging of conformational biosensors that read out distinct active conformations of the δ-opioid receptor (DOR), a physiologically relevant GPCR localized to Golgi and the surface in neuronal cells. We have shown that Golgi and surface pools of DOR both inhibit cAMP, but engage distinct conformational biosensors in response to the same ligand in rat neuroendocrine cells. Further, DOR recruits arrestins on the surface but not on the Golgi. Our results suggest that the local environment determines the active states of receptors for any given drug, allowing GPCRs to couple to different effectors at different subcellular locations.

**\*For correspondence:**
puthenve@umich.edu

**Present address:** [†]Department of Pharmacology and Physiology, University of Rochester Medical Center, Rochester, United States; [‡]Department of Biochemistry and Biophysics, California Institute for Quantitative Biosciences, University of California, San Francisco, San Francisco, United States

**Competing interests:** The authors declare that no competing interests exist.

## Introduction

A given G protein-coupled receptor (GPCR) can generate a diverse array of signaling responses, underscoring the physiological and clinical relevance of this class of proteins. Endogenous and synthetic ligands that 'bias' the responses of a given receptor toward one response or another are a key aspect of this signaling diversity (*Wootten et al., 2018*). This diversity in responses provides several opportunities to target specific GPCR signaling responses to reduce potential adverse effects while managing a variety of clinical conditions. However, using bias to precisely tune GPCR signaling has been difficult, suggesting that we are missing some key pieces in our understanding. Understanding the cellular mechanisms that contribute to individual components of the integrated signaling response is therefore of profound importance to understanding GPCR pharmacology.

The specific conformations adopted by GPCRs, which preferentially allow coupling to distinct effectors, are likely key determinants of which specific downstream signaling response is amplified (*Latorraca et al., 2017*; *Wingler et al., 2019*; *Suomivuori et al., 2020*; *Okude et al., 2015*). Recent studies support an allosteric model of coupling in which binding to both agonist and G protein stabilizes an active state of the receptor, among a number of states that a given receptor may sample (*Okude et al., 2015*; *Weis and Kobilka, 2018*; *Nygaard et al., 2013*; *Ye et al., 2016*; *Manglik et al., 2015*). In addition to canonical G protein effectors, β-arrestins interact with GPCRs through additional receptor conformations and can serve as scaffolds for kinase signaling pathways (*Gurevich and Gurevich, 2019a*; *Liu et al., 2012*; *Wingler et al., 2020*). However, efforts to develop compounds that stabilize one set of conformations and therefore bias the receptor response to specific pathways have been promising but difficult to translate to in vivo models (*Gillis et al., 2020*; *Luttrell et al., 2015*; *Viscusi et al., 2019*).

One hypothesis that could explain this difficulty is that the same agonist could drive coupling of the same receptor to different core signaling proteins based on the immediate subcellular environment of the receptor. Although an exciting idea with profound implications, this hypothesis has been difficult to test using traditional methods, because receptor signaling readouts are complex and have been difficult to separate based on location.

Here we use the δ-opioid receptor (DOR), a physiologically and clinically relevant GPCR, as a model to test this hypothesis. DOR localizes primarily to the Golgi in neuronal cells, with a small amount on the plasma membrane (PM) (*Kim and von Zastrow, 2003*; *Shiwarski et al., 2017a*; *Zhang et al., 1998*; *Cahill et al., 2001a*; *Wang and Pickel, 2001*; *Mittal et al., 2013*; *Shiwarski et al., 2017b*). DOR can be activated both on the PM and the Golgi by synthetic agonists, but whether the two activation states are different is not clear (*Stoeber et al., 2018*). Relocating DOR from intracellular compartments to the PM increases the ability of DOR agonists to relieve pain (*Shiwarski et al., 2017a*; *Mittal et al., 2013*; *Cahill et al., 2001b*; *Patwardhan et al., 2005*), illustrating the importance of understanding whether DOR activation on the Golgi is different from that on the PM.

Here we leverage conformational biosensors and high-resolution imaging to test whether DOR activation on the Golgi is different from that on the PM. We have shown that DOR on the PM, when activated by the selective DOR agonist SNC80, can recruit both a nanobody-based sensor and a G protein-based sensor that read out active DOR conformations, as well as β-arrestins. In contrast, DOR in the Golgi apparatus, when activated by the same ligand, recruits the nanobody sensor, but not the G protein-based sensor or β-arrestins. Nevertheless, Golgi-localized DOR is competent to inhibit cAMP. Together, these data demonstrate that these biosensors could be used to read out subtle differences in GPCR conformations even if signaling readouts are similar. Our results that the downstream effectors recruited by the same GPCR, activated by the same ligand, depend on the location of the receptor, suggest that subcellular location could be a master regulator of GPCR coupling to specific effectors and signaling for any given GPCR-ligand pair.

## Results

Nanobody and miniG protein biosensors are emerging as powerful tools to study the effects of ligand-induced receptor conformational changes at the molecular and cellular level (*Manglik et al., 2017*; *Crilly and Puthenveedu, 2020*; *Wan et al., 2018*). These sensors differentially engage the μ-opioid receptor (MOR) and the κ-opioid receptor (KOR) activated by different agonists in vitro (*Livingston et al., 2018*) or in the PM (*Stoeber et al., 2020*), suggesting that they can provide a readout of conformational heterogeneity in agonist-stabilized active states. Because DOR is highly similar to MOR and KOR in the intracellular regions recognized by the sensors (*Chen et al., 1993*), we asked whether these sensors could be optimized to read out specific active DOR conformations at distinct subcellular locations. We used the nanobody biosensor Nb39, which recognizes opioid receptor active conformations through residues conserved across MOR, KOR, and DOR (*Che et al., 2018*; *Huang et al., 2015*), and the miniGsi biosensor, which mimics the interaction of the Gαi protein with GPCRs (*Wan et al., 2018*; *Nehmé et al., 2017*), as two orthogonal readouts of DOR conformations.

As an initial step, we first tested whether these sensors report active DOR conformations on the PM, similar to what has been reported for MOR and KOR. We used total internal fluorescence reflection microscopy (TIR-FM), which uses an evanescent wave to specifically excite fluorescent proteins on the PM to a depth of approximately 100 nm into the cell (*Hellen and Axelrod, 1991*), to visualize sensor recruitment to activated DOR on the PM with high sensitivity (*Figure 1A*). When cells were treated with either small molecule agonist SNC80 or peptide DPDPE, Nb39 (*Figure 1B–C*) and miniGsi (*Figure 1D–E*) were rapidly recruited to DOR on the plasma membrane (PM DOR), as observed by a rapid increase in fluorescence. Fluorescence of both sensors increased significantly after treatment with either agonist, but not inverse agonist ICI174864 (ICI) (*Figure 1F*), indicating specificity of both sensors for an agonist-induced active conformation. Furthermore, recruitment of Nb39 and miniGsi to PM DOR was concentration-dependent and saturated at the concentration of SNC80 used in these experiments (10 μM) (*Figure 1G*). Concentration-response curves also revealed that miniGsi was more potently recruited to PM DOR than Nb39 in response to the agonist SNC80.

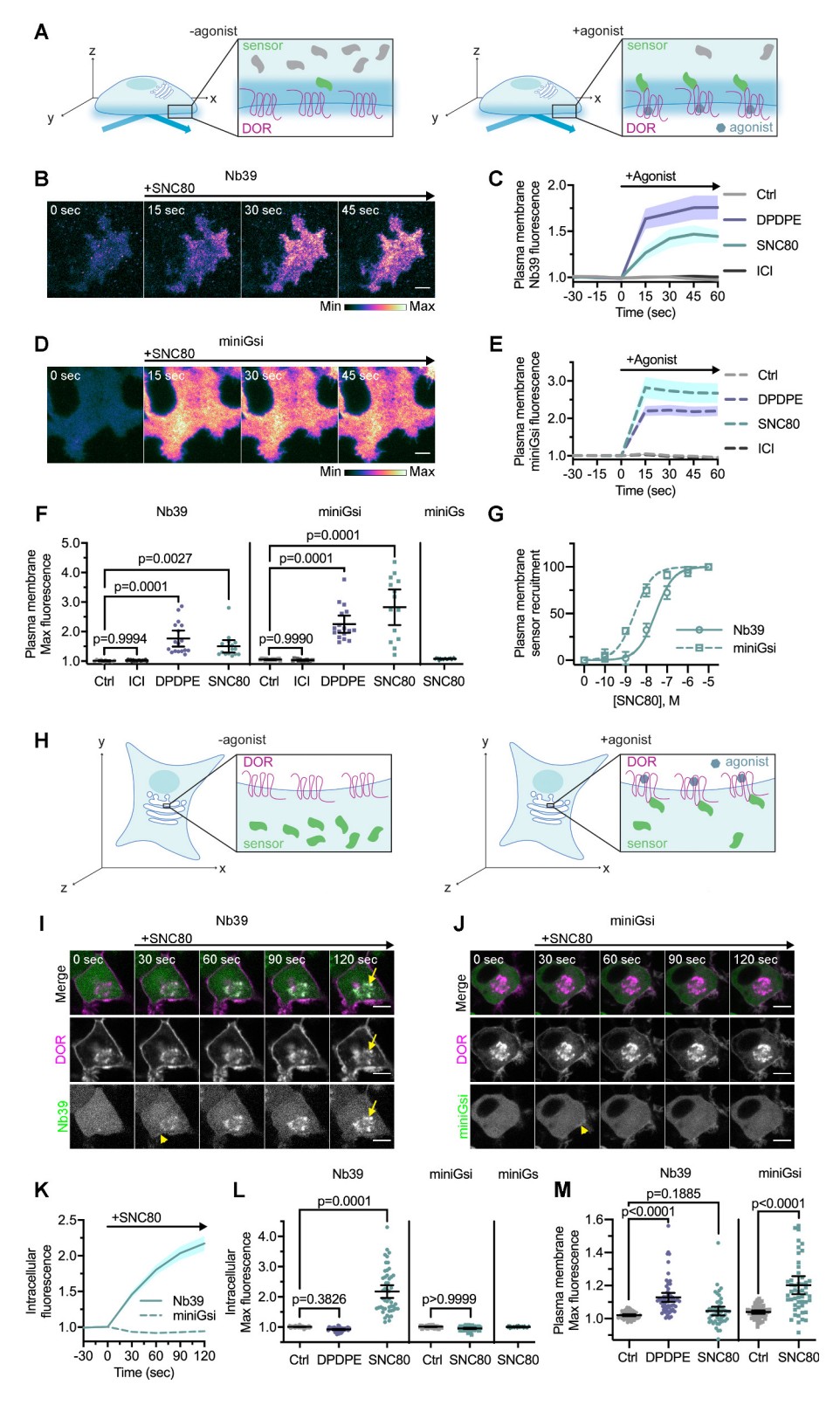

**Figure 1.** Nb39 and miniGsi are differentially recruited to plasma membrane and intracellular DOR. (**A**) Schematic of biosensor recruitment to δ-opioid receptor (DOR) in the plasma membrane (PM) using total internal reflection fluorescence microscopy (TIR-FM). Only fluorescent proteins within the evanescent wave close to the PM were excited, such that baseline fluorescence was low when biosensors were diffuse in the cell but increased upon agonist addition as biosensors were recruited to active DOR in the plasma membrane. (**B**) Nb39-mVenus in PC12 cells expressing SNAP-DOR imaged

*Figure 1 continued on next page*

Figure 1 continued

using TIR-FM to capture recruitment to the PM after addition of 10 µM SNC80 (scale bar = 5 µm). (C) Increase in Nb39-mVenus fluorescence by TIR-FM normalized to the mean baseline fluorescence over time after addition of 10 µM DOR agonist, DPDPE or SNC80, 10 µM inverse agonist, ICI174864 (ICI), or vehicle control (Ctrl, n = 10 cells; ICI, n = 17 cells; DPDPE, n = 17 cells; SNC80, n = 16 cells; all across three biological replicates defined as coverslips prepared and imaged independently; solid line indicates mean, shading ± SEM). (D) Venus-miniGsi in PC12 cells expressing SNAP-DOR imaged using TIR-FM to capture recruitment to the PM after addition of 10 µM SNC80 (scale bar = 5 µm). Calibration bars indicate relative fluorescence values in scaled images. (E) Increase in Venus-miniGsi fluorescence by TIR-FM normalized to the mean baseline fluorescence over time after addition of 10 µM DPDPE or SNC80, 10 µM inverse agonist ICI, or vehicle control (Ctrl, n = 17 cells; ICI, n = 15 cells; DPDPE, n = 17 cells; SNC80, n = 14 cells; all across three biological replicates; dashed line indicates mean, shading ± SEM). (F) Nb39 max PM biosensor fluorescence significantly increased over baseline within 60 s of addition of either agonist DPDPE or SNC80 but not with addition of inverse agonist ICI, by one-way ANOVA (p<0.0001) with p-values from Dunnett's multiple comparisons test to vehicle control reported in the figure. miniGsi max PM biosensor fluorescence significantly increased over baseline within 60 s of addition of either agonist DPDPE or SNC80 but not with addition of inverse agonist ICI, by one-way ANOVA (p<0.0001) with p-values from Dunnett's multiple comparisons test to vehicle control reported in the figure. Venus-miniGs, a sensor for Gs coupling, fluorescence did not visibly increase after addition of 10 µM SNC80 (Nb39: Ctrl, n = 10 cells; ICI, n = 17 cells; DPDPE, n = 17 cells; SNC80, n = 16 cells; miniGsi: Ctrl, n = 17 cells; ICI, n = 15 cells; DPDPE, n = 17 cells; SNC80, n = 14 cells; miniGs-SNC80, n = 20 cells; all across three biological replicates; mean ± 95% CI, points represent individual cells). (G) Concentration-response curves for Nb39 ($EC_{50}$ = 22.7 nM) and miniGsi ($EC_{50}$ = 2.284 nM) plasma membrane recruitment measured in TIR-FM, in cells treated with increasing concentrations of SNC80 ranging from 0.1 nM to 10 µM. Responses were normalized from 0 to 100 for cells within each condition (Nb39, n = 13 cells; miniGsi, n = 7 cells; symbols indicate mean normalized response for cells in each condition with error bars indicating ± 95% CI; solid and dashed lines indicate fitted non-linear curves with a standard slope of 1, for Nb39 and miniGsi, respectively). (H) Schematic of biosensor recruitment to DOR in intracellular compartments upon addition of a cell-permeable agonist. Both Nb39 and miniGsi biosensors were diffuse throughout the cytoplasm in the absence of agonist (left), but were expected to localize to membranes containing active receptor upon agonist addition (right). (I) PC12 cells expressing SNAP-DOR (magenta in merge) and Nb39-mVenus (green in merge) were treated with 10 µM SNC80 and imaged live by confocal microscopy. Treatment with SNC80 led to an increase in Nb39-mVenus signal in a perinuclear region (yellow arrow), which colocalized with intracellular DOR (white in merge). A small amount of Nb39 recruitment is also visible at the PM (yellow arrowhead) (scale bar = 5 µm). (J) PC12 cells expressing SNAP-DOR (magenta in merge) and Venus-miniGsi (green in merge) were treated with 10 µM SNC80 and imaged live by confocal microscopy. miniGsi did not localize to intracellular DOR after agonist treatment, though a small amount of miniGsi recruitment is visible at the PM (yellow arrowhead) (scale bar = 5 µm). (K) Nb39 (solid line indicates mean, shading ± SEM) and miniGsi (dashed line, shading ± SEM) fluorescence in the region of the cell defined by intracellular DOR normalized to mean baseline fluorescence over time after addition of 10 µM SNC80 (Nb39, n = 49 cells across four biological replicates; miniGsi, n = 51 cells across three biological replicates). (L) Max intracellular biosensor fluorescence in the region of the cell defined by intracellular DOR within 120 s of agonist addition shows a significant increase in Nb39 recruitment with addition of permeable agonist SNC80 but not with peptide agonist DPDPE, by one-way ANOVA (p<0.0001) with p-values from Dunnett's multiple comparisons test to vehicle control reported in the figure. In contrast, miniGsi intracellular max fluorescence did not increase upon addition of 10 µM SNC80 by one-tailed Student's t-test compared to vehicle control. miniGs intracellular max fluorescence also did not visibly increase upon SNC80 treatment (Nb39: Ctrl, n = 61 cells; DPDPE, n = 61 cells; SNC80, n = 49 cells; miniGsi: Ctrl, n = 57 cells; SNC80, n = 51 cells; miniGs: SNC80, n = 36 cells; all across a minimum of three biological replicates; mean ± 95% CI, points represent individual cells). (M) Max plasma membrane biosensor fluorescence in the region of the cell defined by plasma membrane DOR for the same cells quantified in (L) shows a significant increase in Nb39 recruitment with addition of DPDPE and a small but non-significant increase upon addition of SNC80, by one-way ANOVA (p<0.0001). P-values from Dunnett's multiple comparisons test to vehicle control are reported. MiniGsi plasma membrane max fluorescence also significantly increased upon addition of SNC80, as estimated by one-tailed Student's t-test, compared to vehicle control (Nb39: Ctrl, n = 61 cells; DPDPE, n = 61 cells; SNC80, n = 49 cells; miniGsi: Ctrl, n = 57 cells; SNC80, n = 51 cells; miniGs: SNC80, n = 36 cells; all across a minimum of three biological replicates; mean ± 95% CI, points represent individual cells with one outlier in the miniGsi SNC80 condition equal to 2.0167, not shown in the graph).

The online version of this article includes the following figure supplement(s) for figure 1:

**Figure supplement 1.** DOR expression levels are similar across treatment conditions.

**Figure supplement 2.** Nb39 recruitment to active DOR is reversible.

**Figure supplement 3.** Mechanism of DOR Golgi retention does not influence sensor recruitment to Golgi or PM DOR, and sensor recruitment to intracellular DOR is not correlated with sensor expression.

Interestingly, DPDPE recruited Nb39 to PM DOR more strongly than SNC80 (*Figure 1C,F*), whereas the opposite trend was observed for miniGsi (*Figure 1E–F*), suggesting that these sensors might report selective conformations of DOR induced by different agonists. MiniGs, which mimics Gαs protein interaction with Gs-coupled GPCRs, was not recruited to PM DOR (*Figure 1F*), suggesting that the miniGsi sensor specifically reports activation of the Gi-coupled DOR. DOR expression levels were, overall, comparable across conditions (*Figure 1—figure supplement 1A*). Overall, our data show that Nb39 and miniGsi report agonist-induced active conformations of PM DOR.

To test whether DOR localized to an intracellular compartment engages Nb39 or miniGsi differently upon activation by the same agonist, we took advantage of the fact that newly synthesized DOR is retained in an intracellular compartment in neurons and pheochromocytoma-12 cells (PC12 cells) (*Figure 1H and I*, yellow arrows) acutely treated with nerve growth factor (NGF) (*Kim and von*

*Zastrow, 2003*; *Shiwarski et al., 2017a*). The presence of both plasma membrane and intracellular pools of DOR in these cells allowed us to measure sensor recruitment to DOR at both of these locations.

We first tested whether the two different biosensors were differentially recruited to DOR in intracellular compartments (IC DOR) vs PM DOR by confocal imaging (*Figure 1H*). When cells were treated with 10 μM SNC80, a membrane-permeable, small molecule agonist, Nb39 was rapidly recruited to intracellular SNAP-tagged DOR, within 30 s (*Figure 1I*, *Video 1*). When quantitated, Nb39 fluorescence in the region of the cell defined by IC DOR rapidly and significantly increased after SNC80 addition (*Figure 1K–L*). This Nb39 recruitment was dynamic and required DOR activation, as the DOR antagonist naltrindole rapidly reversed this effect (*Figure 1—figure supplement 2A–B*). In striking contrast to Nb39, miniGsi was not recruited to IC DOR in the same time frame (*Figure 1J–K*, *Video 2*), despite comparable levels of IC DOR (*Figure 1—figure supplement 1B*). MiniGsi fluorescence in the region of the cell defined by IC DOR did not increase in cells treated with SNC80 (*Figure 1K–L*). As a control, miniGs was also not recruited to IC DOR in cells treated with SNC80 (*Figure 1L*).

Both sensors were recruited to PM DOR, as seen by confocal imaging in the same cells in which we measured recruitment to IC DOR, although the sensitivity of detection was lower in confocal imaging (*Figure 1I–J*, yellow arrowheads, and *Figure 1M*). MiniGsi was recruited to PM DOR more strongly than Nb39 (*Figure 1F,M*), consistent with the TIR-FM results, which makes the absence of miniGsi recruitment to IC DOR in response to SNC80 even more striking. These results indicate that SNC80 promotes an active DOR conformation preferentially recognized by Nb39 in intracellular compartments and a distinct active DOR conformation preferentially recognized by miniGsi at the PM, suggesting location-specific conformational effects of SNC80 on DOR.

Differential biosensor recruitment to IC DOR did not depend on the method used to cause DOR retention in this compartment (*Figure 1—figure supplement 3A*), and recruitment to PM DOR was unaffected by the presence of IC DOR (*Figure 1—figure supplement 3B*). Further, recruitment of either sensor was not significantly correlated with sensor expression level (*Figure 1—figure supplement 3C–D*), as miniGsi failed to show recruitment to IC DOR across a broad range of expression levels (*Figure 1—figure supplement 3D*).

When cells were treated with 10 μM peptide agonist DPDPE, which does not readily cross the PM over short time scales (*Stoeber et al., 2018*), Nb39 was not recruited to IC DOR (*Figure 1L*). This suggests that activation of PM DOR is not sufficient for recruitment of Nb39 to IC DOR. We next tested whether PM DOR activation was required for differential sensor recruitment to IC DOR. Cells were pretreated with a high concentration (100 μM) of DOR inverse agonist ICI, a peptide restricted to the extracellular space (*Stoeber et al., 2018*), to pharmacologically block PM DOR. Nb39 recruitment to IC DOR after 100 nM SNC80 addition was then measured. Nb39 was robustly recruited to IC DOR, even when PM DOR was pharmacologically blocked, indicating that recruitment to IC DOR does not require activation of PM DOR (*Figure 2A*, top, 2B, C). When PM DOR was pharmacologically blocked, miniGsi again remained diffuse throughout the cell and was not recruited to IC DOR (*Figure 2A*, bottom, 2B, C), indicating that the absence of miniGsi recruitment to IC DOR is not due to sequestration of sensor at PM DOR. To test whether activation of endogenous G proteins was restricting miniGsi recruitment, cells were pretreated with pertussis toxin (PTX) to inactivate endogenous Gαi/o proteins. Even in PTX-treated cells, miniGsi was not recruited to IC DOR, suggesting that competition with endogenous Gαi/o protein effectors for interaction with DOR is not responsible for the lack of recruitment of miniGsi to IC DOR (*Figure 2C*).

Immunofluorescence microscopy showed that Nb39 was recruited to IC DOR localized to the

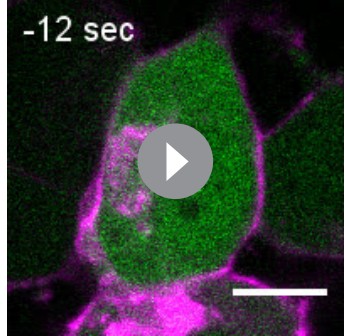

**Video 1.** PC12 cells expressing SNAP-DOR (magenta) and Nb39 (green) imaged live after addition of 10 μM SNC80 (scale bar = 5 μm). Nb39 was rapidly recruited to intracellular δ-opioid receptor (DOR) (seen as white due to colocalization) upon SNC80 addition.
https://elifesciences.org/articles/67478#video1

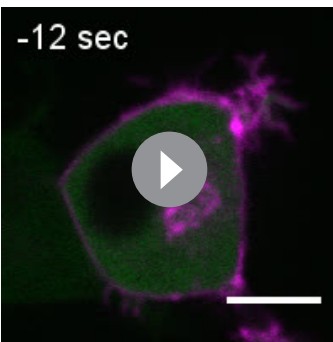

**Video 2.** PC12 cells expressing SNAP-DOR (magenta) and miniGsi (green) imaged live after addition of 10 μM SNC80 (scale bar = 5 μm). In contrast to Nb39, miniGsi was not recruited to intracellular δ-opioid receptor (DOR) upon SNC80 addition.
https://elifesciences.org/articles/67478#video2

Golgi. Using a similar approach as described above, PC12 cells expressing Flag-tagged DOR and either Nb39 or miniGsi were pretreated for 15 min with 10 μM β-chlornaltrexamine (CNA), an irreversible, cell-impermeable antagonist (*Shiwarski et al., 2017a*; *Virk and Williams, 2008*), to irreversibly block PM DOR, before treating with 10 μM SNC80 for 5 min. Cells were stained for TGN-38, a marker for the trans-Golgi network, which was previously shown to colocalize with IC DOR (*Shiwarski et al., 2017a*). Consistent with live-cell imaging data, only Nb39, and not miniGsi, was recruited to IC DOR in a region of the cell colocalizing with the TGN-38 marker (*Figure 2D,E*) in cells treated with CNA and SNC80. CNA alone did not cause recruitment of either sensor. Sensor fluorescence in the region of the cell defined by TGN-38 staining was normalized to sensor fluorescence in the cell outside this region, as a measure of sensor enrichment in the Golgi. Treatment with CNA and SNC80 significantly increased Nb39 Golgi enrichment (*Figure 2F*), whereas miniGsi enrichment was not significantly different from control cells (*Figure 2G*). These results confirm differential biosensor recruitment to IC DOR specifically localized to the Golgi and reiterate that PM DOR activation is not required for differential biosensor recruitment to Golgi DOR.

In addition to heterotrimeric G proteins, DOR and other GPCRs interact with other proteins and signaling effectors after agonist-induced conformational changes. Agonist-dependent differential biosensor recruitment to MOR and KOR in the PM correlates with recruitment of other receptor effectors, specifically G protein-coupled receptor kinase 2 (GRK2), which mediates receptor desensitization (*Stoeber et al., 2020*). Given differential recruitment of Nb39 and miniGsi to IC DOR, we hypothesized that Golgi localization may also influence coupling to other downstream signaling effectors. β-arrestins interact with DOR and other GPCRs after activation by agonists and receptor phosphorylation by GRKs to mediate receptor desensitization and internalization from the PM (*Zhang et al., 1999*; *Zhang et al., 2005*; *Gurevich and Gurevich, 2019b*). β-arrestins can also scaffold kinase-signaling complexes from GPCRs at the PM and endosomes (*Peterson and Luttrell, 2017*; *DeFea et al., 2000a*; *DeFea et al., 2000b*; *McDonald et al., 2000*; *Weinberg et al., 2017*).

To test whether β-arrestin effectors are recruited to active IC DOR, we monitored recruitment of fluorescently tagged β-arrestin-1 or β-arrestin-2 to IC DOR. Similar to miniGsi, neither β-arrestin-1 nor β-arrestin-2 was recruited to IC DOR in cells treated with SNC80, and no increase in fluorescence in the region of the cell defined by IC DOR was detected (*Figure 3A–C*). In contrast, both β-arrestins were visibly recruited to the PM by confocal imaging (*Figure 3A,B*, yellow arrowheads) and by quantitation of TIR-FM imaging (*Figure 3D–F*) in response to SNC80 treatment. Together, these data indicate that like miniGsi, β-arrestins interact with only agonist-activated DOR present in the PM.

Given the differential recruitment of active conformation biosensors, Nb39 and miniGsi, and the absence of β-arrestin recruitment to IC DOR, we asked whether the active conformation of IC DOR allows for signaling through G proteins. Like the other opioid receptors, DOR couples primarily to Gαi/o proteins, which inhibit adenylyl cyclase activity to decrease cAMP (*Gendron et al., 2016*). We used a Forster resonance energy transfer (FRET) sensor, ICUE3, to monitor cAMP levels in single cells in real time (*DiPilato and Zhang, 2009*). In PC12 cells expressing ICUE3 and SNAP-DOR, addition of adenylyl cyclase activator forskolin (Fsk, 2 μM) caused a rapid increase in the cyan fluorescent protein (CFP/FRET) ratio over the baseline ratio (*Figure 4A–B*). Pretreatment with 100 nM SNC80 prior to Fsk addition decreased the Fsk-stimulated cAMP response (*Figure 4C–D*), consistent with DOR activation inhibiting adenylyl cyclase activity.

We next specifically tested whether IC DOR was sufficient for cAMP inhibition. Cells expressing DOR were treated with cycloheximide (CHX) to ensure that, in the absence of NGF, newly synthesized DOR transiting the Golgi was cleared out and that no residual DOR remained in the Golgi. In cells pretreated with NGF before CHX, the pool of IC DOR was maintained even after CHX

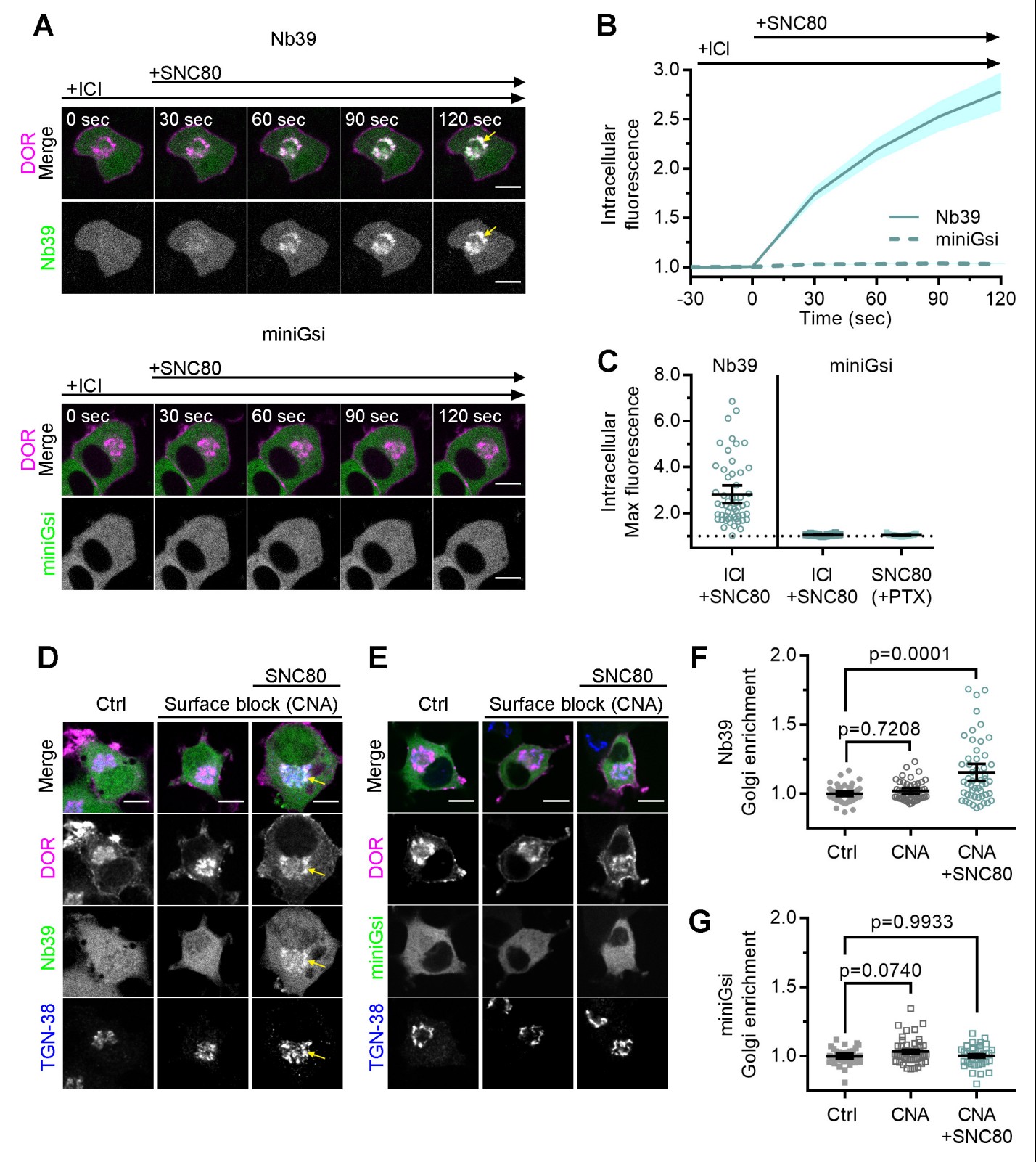

**Figure 2.** Differential sensor recruitment to Golgi DOR is independent of plasma membrane DOR activation. (**A**) PC12 cells expressing SNAP-DOR (magenta in merge) and Nb39-mVenus or Venus-miniGsi (green in merge) were imaged live by confocal microscopy with 100 μM ICl174864 (ICI) present in the media before addition of 100 nM SNC80. After SNC80 treatment, Nb39-mVenus fluorescence increased in a perinuclear region (yellow arrow), which colocalized with intracellular δ-opioid receptor (DOR) (white in merge), whereas Venus-miniGsi remained diffuse through the cell (scale bar = 5

*Figure 2 continued on next page*

*Figure 2 continued*

µm). (**B**) Nb39 (solid line indicates mean, shading ± SEM) or miniGsi (dashed line, shading ± SEM) fluorescence in the region of the cell defined by intracellular DOR normalized to mean baseline fluorescence in cells treated with 100 µM ICI and 100 nM SNC80 (Nb39, n = 54 cells across three biological replicates; miniGsi, n = 58 cells across four biological replicates). (**C**) Nb39-mVenus max intracellular fluorescence increased over baseline within 120 s of SNC80 in cells treated with 100 µM ICI and 100 nM SNC80. In contrast, miniGsi intracellular max fluorescence did not visibly increase over baseline in cells treated with ICI and SNC80, nor in cells pretreated with pertussis toxin (PTX) and SNC80 (Nb39: ICI + SNC80, n = 54 cells; miniGsi: ICI + SNC80, n = 58 cells; PTX + SNC80, n = 33 cells; mean ± 95% CI, points represent individual cells). (**D**) PC12 cells expressing Flag-DOR and Nb39-mVenus or Venus-miniGsi, (**E**) (green in merge) were treated with either 10 µM β-chlornaltrexamine (CNA) alone for 15 min or 10 µM CNA for 15 min followed by 10 µM SNC80 for 5 min, then fixed and stained for Flag (magenta in merge) and trans-Golgi network marker TGN-38 (blue in merge) (scale bar = 5 µm). Colocalization of DOR, Nb39, and TGN-38 is visible in white and light blue (yellow arrow) in cells treated with CNA and SNC80, but not CNA alone. (**F**) Normalized Nb39-mVenus fluorescence enriched in the Golgi, expressed as sensor fluorescence in the region of the cell defined by TGN-38 staining divided by sensor fluorescence in the region of the cell not containing TGN-38 staining. Nb39 Golgi enrichment was significantly increased in cells treated with CNA and SNC80, but not CNA alone, by one-way ANOVA (p<0.0001) with p-values reported in the figure from Dunnett's multiple comparisons test compared to control cells (Ctrl, n = 46 cells; CNA, n = 49; CNA + SNC80, n = 52; all across two biological replicates; points indicate individual cells with bars representing mean ± 95% CI). (**G**) Venus-miniGsi Golgi enrichment was not significantly increased in cells treated with either CNA and SNC80 or CNA alone, by one-way ANOVA (p=0.0654) with p-values reported in the figure from Dunnett's multiple comparisons test compared to control cells (Ctrl, n = 40 cells; CNA, n = 50; CNA + SNC80, n = 37; all across two biological replicates; points indicate individual cells with bars representing mean ± 95% CI).

treatment, consistent with previous results (*Kim and von Zastrow, 2003*; *Shiwarski et al., 2017a*). Under these conditions, the overall Fsk response and SNC80-mediated inhibition in cells treated with NGF were comparable to untreated cells (*Figure 4—figure supplement 1A*). To isolate the contribution of IC DOR to cAMP inhibition, we pharmacologically blocked PM DOR with 100 µM ICI. In cells with PM DOR only, SNC80 failed to decrease Fsk-stimulated cAMP in the presence of ICI (*Figure 4E–F*). Neither the total cAMP levels, measured as area under the curve, nor the endpoint cAMP levels, measured as the change in endpoint cAMP levels over baseline, were significantly different from cells treated with Fsk alone (*Figure 4I–J*). In contrast, in cells with IC DOR, SNC80 decreased Fsk-stimulated cAMP even in the presence of ICI (*Figure 4G–H*), and significantly decreased endpoint and total cAMP levels (*Figure 4I–J*). IC DOR activation suppressed Fsk-stimulated cAMP to approximately half the degree suppressed by combined PM and IC DOR (*Figure 4I–J*, *Figure 4—figure supplement 1A*). As a control, ICI alone did not significantly affect endpoint or total cAMP levels (*Figure 4I–J*). These results indicate that Golgi DOR activation is sufficient for cAMP inhibition.

As an independent method to induce an intracellular pool of DOR, we used LY294002, a small molecule inhibitor of PI3K that causes DOR retention in the Golgi independent of NGF (*Shiwarski et al., 2017b*). Similar to results obtained in NGF-treated cells, the permeable small molecule agonist SNC80 decreased Fsk-stimulated cAMP (*Figure 4—figure supplement 1B–D*) even in the presence of ICI, reiterating that Golgi DOR activation is sufficient for cAMP inhibition. The total and endpoint cAMP levels decreased significantly compared to cells treated with Fsk alone (*Figure 4—figure supplement 1G–H*). Again, IC DOR alone suppressed Fsk-stimulated cAMP to approximately half the degree suppressed by combined PM and IC DOR (*Figure 4—figure supplement 1B,G–H*). As a control, the peptide DPDPE agonist did not decrease the Fsk response in the presence of ICI (*Figure 4—figure supplement 1B,E–F*).

## Discussion

Together, our data suggest that DOR activation by the same agonist in different subcellular compartments promotes distinct active conformations recognized differentially by biosensors. The conventional model suggests that distinct GPCR conformations can drive coupling to distinct effectors, which determine subsequent downstream signaling responses. This relationship between structure and function has been a subject of great interest due to its potential to explain the pleiotropic effects of GPCR activation by any given ligand. Our results suggest that the subcellular location in which receptors are activated might determine the conformational landscapes that receptors can adopt upon activation, because the immediate environment of receptors varies between these locations.

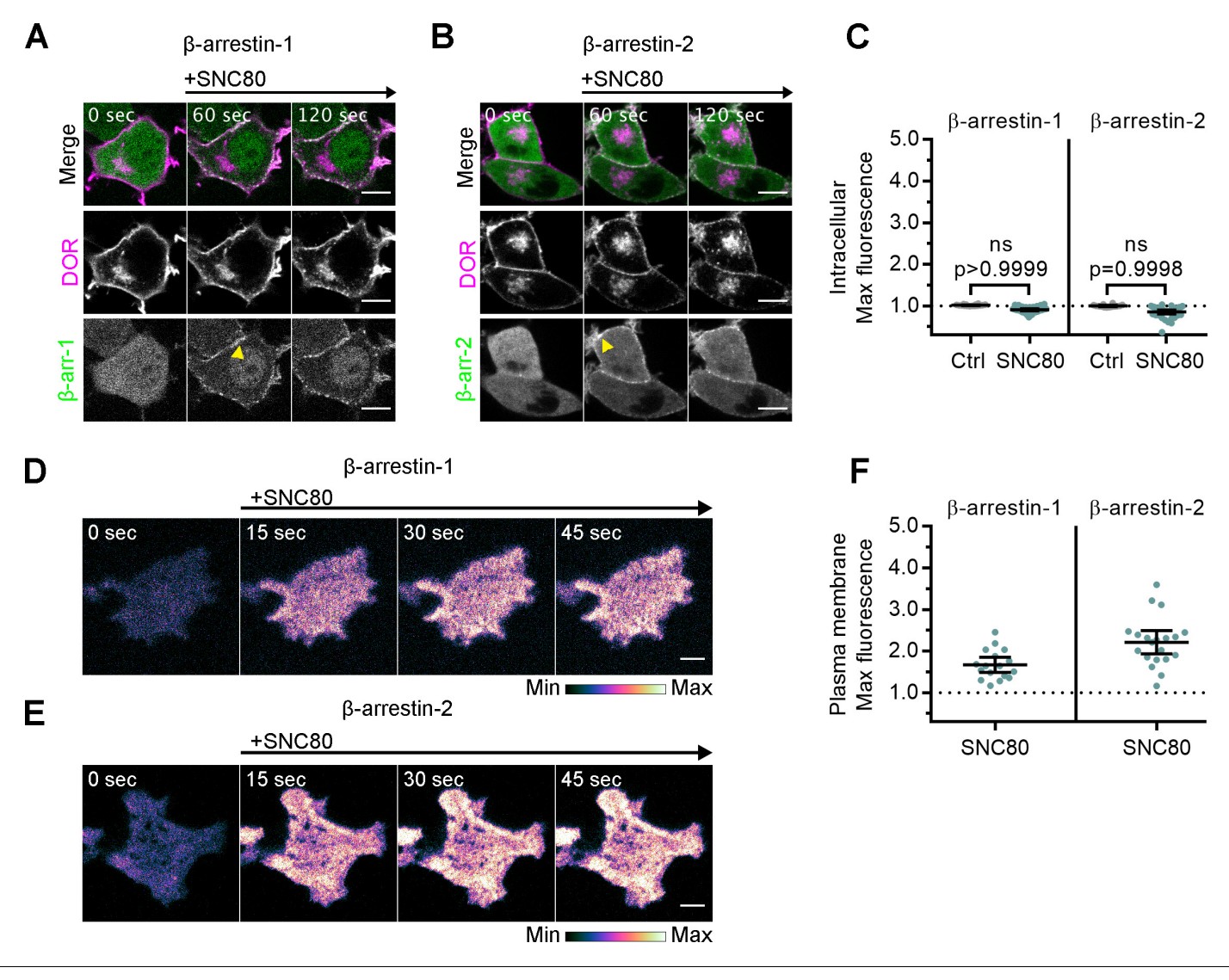

**Figure 3.** Arrestins are differentially recruited to plasma membrane and intracellular DOR. (**A**) PC12 cells expressing SNAP-DOR (magenta in merge) and β-arrestin-1-mScarlet (green in merge) were treated with 10 μM SNC80 and imaged live by confocal microscopy. β-arrestin-1-mScarlet signal increased at the plasma membrane (PM) (yellow arrowhead) but not at sites colocalized with intracellular δ-opioid receptor (DOR) upon 10 μM SNC80 treatment (scale bar = 5 μm). (**B**) PC12 cells expressing SNAP-DOR (magenta in merge) and β-arrestin-2-tdTomato (green in merge). β-arrestin-2-tdTomato signal increased at the PM (yellow arrowhead) but not at sites colocalized with intracellular DOR upon 10 μM SNC80 treatment (scale bar = 5 μm). (**C**) Neither β-arrestin-1-mScarlet nor β-arrestin-2-tdTomato max intracellular fluorescence significantly increased within 120 s of SNC80 addition by one-tailed Student's t-test compared to control cells (β-arr-1: Ctrl, n = 16 cells; SNC80, n = 33 cells; β-arr-2: Ctrl, n = 14 cells; SNC80, n = 37 cells; with control conditions across one biological replicate and SNC80 conditions across three biological replicates; mean ± 95% CI, points represent individual cells). (**D**) β-arrestin-1-mScarlet in PC12 cells expressing SNAP-DOR imaged using total internal fluorescence reflection microscopy (TIR-FM) to capture recruitment to the PM after addition of 10 μM SNC80 (scale bar = 5 μm). (**E**) β-arrestin-2-tdTomato in PC12 cells expressing SNAP-DOR imaged using TIR-FM to capture recruitment to the PM after addition of 10 μM SNC80 (scale bar = 5 μm). Calibration bars indicate relative fluorescence values in scaled images. (**F**) Both β-arrestin-1-mScarlet and β-arrestin-2-tdTomato max PM fluorescence increased within 60 s of 10 μM SNC80 addition (β-arr-1: SNC80, n = 17 cells; β-arr-2: SNC80, n = 20 cells; all across three biological replicates; mean ± 95% CI, points represent individual cells).

Conformational biosensors like Nb39 and miniGsi used here are valuable tools to study how location can bias receptor conformations. The possibility that Nb39 and miniGsi could recognize distinct conformations is supported by structures of agonist-bound homologous MOR and KOR in complex with Nb39 or MOR in complex with the heterotrimeric G protein complex, $G\alpha_{i1}\beta_1\gamma_2$. Structures of MOR with agonist BU72 and KOR with agonist MP1104 share the outward shift of transmembrane helix (TM)6, which is characteristic of active GPCR structures (*Che et al., 2018*; *Huang et al., 2015*).

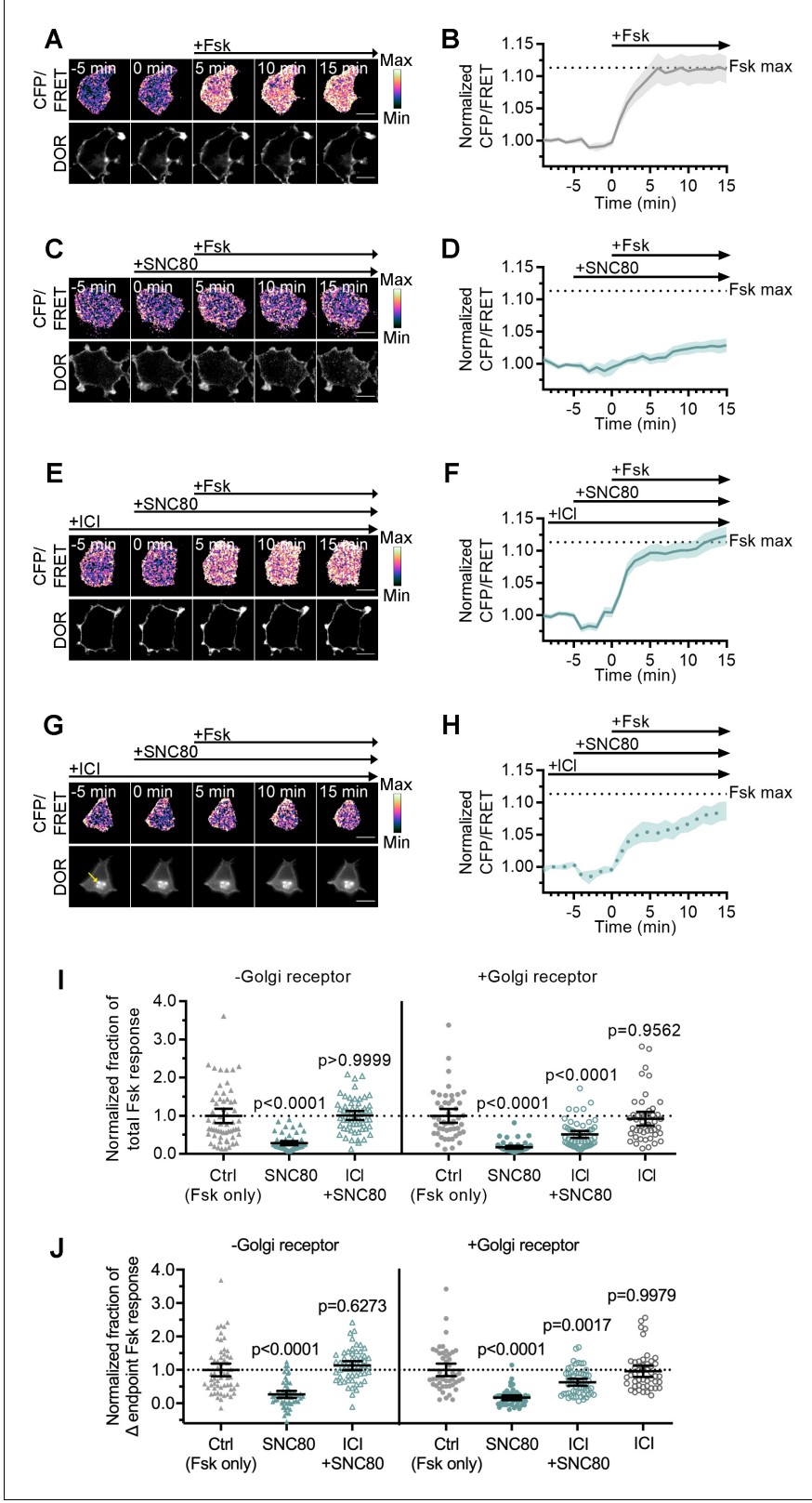

**Figure 4.** Golgi DOR inhibits cAMP. **(A–H)** Ratiometric CFP/Forster resonance energy transfer (FRET) and receptor images, along with corresponding trace of mean cellular CFP/FRET ratios (solid line indicates mean, shading ± 95% CI), in PC12 cells expressing the ICUE3 cAMP FRET sensor and SNAP-DOR (scale bar = 10 µm). Calibration bars indicate relative fluorescence values in scaled images. **(A–B)** In cells without intracellular δ-opioid receptor (DOR), CFP/FRET ratio increased over baseline upon treatment with 2 µM forskolin (Fsk), consistent with increase in cellular cAMP levels. **(C–D)**

*Figure 4 continued on next page*

*Figure 4 continued*

Treatment with DOR agonist SNC80 (100 nM) decreased Fsk-stimulated increase in CFP/FRET ratio. (E–F) In cells without intracellular DOR, SNC80-dependent decrease in Fsk-stimulated CFP/FRET ratio was reversed when peptide inverse agonist ICI174864 (ICI) (100 μM) was present in media. (G–H) In cells containing intracellular DOR (G, yellow arrow), SNC80 decreased Fsk-stimulated CFP/FRET ratio even when ICI was present in media. (I–J) Fsk-stimulated total cAMP levels (area under the curve) (I) and endpoint CFP/FRET ratios (J), normalized to mean of control treated cells within -Golgi receptor and +Golgi receptor groups. Treatment with 100 nM SNC80 significantly decreased total Fsk-stimulated cAMP and endpoint ratios. ICI and SNC80 treatment of cells without Golgi DOR did not significantly decrease total cAMP or endpoint ratios. In contrast, ICI and SNC80 treatment of cells with Golgi DOR significantly decreased total cAMP and endpoint ratios. ICI treatment alone in cells with Golgi DOR did not significantly decrease total cAMP and endpoint ratios (-Golgi receptor: control, n = 59 cells; SNC80, n = 58; ICI + SNC80, n = 57; +Golgi DOR: control, n = 48; SNC80, n = 50; ICI + SNC80, n = 55; ICI, n = 47; all across two biological replicates; one-way ANOVA (total cAMP, p<0.0001; endpoint cAMP, p<0.0001) with p-values reported in the figure from Sidak's multiple comparisons test for each condition compared to control cells within –Golgi receptor and +Golgi receptor groups).

The online version of this article includes the following figure supplement(s) for figure 4:

**Figure supplement 1.** Golgi DOR inhibits cellular cAMP.

Nb39 appears to stabilize this conformation via contacts with intracellular loop (ICL)2 and ICL3, as well as the eighth helix through residues conserved across MOR, KOR, and DOR (*Che et al., 2018*; *Huang et al., 2015*). The structure of MOR in complex with agonist [D-Ala2, N-MePhe4, Gly-ol5]-enkephalin (DAMGO) and the nucleotide-free Gαi protein is very similar to the MOR-Nb39 structure, with the exception of a greater displacement of TM6 toward TM7 and a decreased extension of ICL3 (*Koehl et al., 2018*). The miniGsi sensor does not contain all regions of Gαi that contact the receptor, but many of the residues that interact with MOR ICL2 and ICL3 via the Gαi C-terminal α5 helix are present in miniGsi, and previous reports show that miniGs and Gαs contact B2AR similarly (*Nehmé et al., 2017*; *Carpenter et al., 2016*). Though Nb39 and Gαi contact opioid receptors in similar regions and share two interaction residues, each also makes additional distinct contacts with TM domains and the eighth helix. Distinct interactions with these intracellular domains important for effector coupling and unique stabilization of TM6 and ICL3 by Nb39 could suggest that these sensors differentially report distinct conformations relevant to receptor function. Additionally, these structures are limited to a single static view of opioid receptor-active conformation, and the ability of Nb39 and miniGsi to discriminate between additional distinct intermediate or active conformations will be an exciting area for future study.

One clear difference between compartments is the composition of specific phospholipids that make up the membranes (*Ikonen, 2008*; *van Meer et al., 2008*; *Balla, 2013*). Phospholipids differing in charge can stabilize active or inactive conformations of the β$_2$-adrenergic receptor, stabilize G protein coupling, and modulate G protein selectivity (*Dawaliby et al., 2016*; *Strohman et al., 2019*; *Yen et al., 2018*). Lipid composition can also influence recruitment of effectors like β-arrestins, which bind PI4,5P2, a phospholipid species enriched in the PM (*Gaidarov et al., 1999*), potentially contributing to the lack of observed arrestin recruitment to Golgi DOR. To date, β-arrestin recruitment to active GPCRs in the Golgi has not been reported, and the impact of receptor localization to this compartment on desensitization, β-arrestin recruitment, and β-arrestin-biased signaling is not known.

Other compartment-specific factors including ion concentrations and GPCR-interacting proteins could influence receptor conformations and effector coupling. The Golgi lumen is more acidic than the extracellular space, pH 6.4 vs pH 7.4 (*Kim et al., 1996*; *Llopis et al., 1998*; *Miesenböck et al., 1998*), which could affect ligand binding and GPCR activation (*Ghanouni et al., 2000*; *Meyer et al., 2019*; *Vetter et al., 2006*; *Pert and Snyder, 1973*). Estimated concentrations of sodium in the Golgi are closer to cytosolic sodium concentrations (12–27 mM) than high extracellular sodium concentrations (100 mM) (*Chandra et al., 1991*; *Hooper and Dick, 1976*). Sodium acts as an allosteric modulator of class A GPCRs, and DOR specifically has been crystallized with a coordinated sodium ion, which stabilizes the inactive receptor conformation, and is required for receptor activation and signaling (*Fenalti et al., 2014*; *Zarzycka et al., 2019*), suggesting Golgi sodium concentrations could affect DOR activity. Ligand concentrations of a permeable agonist like SNC80 may also differ between the Golgi lumen and the extracellular space. A lower SNC80 concentration in the Golgi, however, is unlikely to explain the differential recruitment we observed, as miniGsi is more potently recruited to PM DOR than Nb39 (*Figure 1G*). Additionally, DOR-interacting proteins, which regulate

DOR trafficking and localize to the Golgi, like the coatomer protein I (COPI) complex and Rab10, could also regulate receptor conformations and effector coupling (*Degrandmaison et al., 2020*; *St-Louis et al., 2017*; *Shiwarski et al., 2019*).

Our results suggest that the conformational space sampled by any given GPCR, even when activated by the same agonist, differs based on the precise subcellular location of the receptor. Compartmental effects on GPCR conformations may also be specific to individual GPCRs. In contrast to DOR, both an active-state nanobody and miniGs are recruited to active Gαs-coupled β$_1$-adrenergic receptor (B1AR) in the Golgi, suggesting that the local Golgi environment may influence DOR and B1AR energy landscapes differently (*Irannejad et al., 2017*; *Nash et al., 2019*). Additionally, the A$_1$-adenosine receptor, when expressed exogenously, can localize to the Golgi and recruit miniGsi to this compartment upon adenosine treatment (*Wan et al., 2018*). This result demonstrates that mini-Gsi can in fact report active conformations of Gαi-coupled receptors in the Golgi, which emphasizes the absence of miniGsi recruitment to Golgi DOR that we observed. These GPCR-specific effects may reflect important differences in pharmacology among individual GPCRs and emphasize the importance of characterizing compartmental effects for each GPCR.

These results also provide a new perspective into drug development efforts, by highlighting the effects that the subcellular location of receptors could have on the integrated effects of any given drug. The majority of these efforts largely rely on assays using conventional readouts of signaling in model cells, where GPCR localization could be different from that of physiologically relevant cells in vivo. This difference is especially true for DOR, which exhibits robust surface localization in model cell lines, but high levels of intracellular pools in many neuronal subtypes. Traditional signaling assays, which rely on whole-cell readouts of primary signaling pathways such as cAMP, will not distinguish between the contributions of different pools of receptors, which could signal differently via pathways outside the primary readouts (*Costa-Neto et al., 2016*). Therefore, the potentially distinct effects of ligands at spatially distinct pools of receptors in the integrated response should be an important consideration for measuring the outcomes of receptor activation.

# Materials and methods

## Key resources table

| Reagent type (species) or resource | Designation | Source or reference | Identifiers | Additional information |
|---|---|---|---|---|
| Cell line (*Rattus norvegicus*, male) | PC12 | ATCC | CRL-1721 | |
| Recombinant DNA reagent | SNAP-DOR | This paper | | pcDNA3.1 backbone; see 'Materials and methods' |
| Recombinant DNA reagent | SSF-DOR | *Kim and von Zastrow, 2003*; PMID:12657666 | | |
| Recombinant DNA reagent | Nb39-mVenus | *Che et al., 2020*; PMID:32123179 | | |
| Recombinant DNA reagent | Venus-miniGsi | *Wan et al., 2018*; PMID:29523687 | | |
| Recombinant DNA reagent | Venus-miniGs | *Wan et al., 2018*; PMID:29523687 | | |
| Recombinant DNA reagent | β-arrestin-1-mScarlet | This paper | | pcDNA3.1 backbone; see 'Materials and methods' |
| Recombinant DNA reagent | β-arrestin-2-tdTomato | *Weinberg et al., 2017*; PMID:28153854 | | |
| Recombinant DNA reagent | ICUE3 | Addgene; *DiPilato and Zhang, 2009*; PMID:19603118 | Plasmid #61622 | |

*Continued on next page*

*Continued*

| Reagent type (species) or resource | Designation | Source or reference | Identifiers | Additional information |
|---|---|---|---|---|
| Antibody | Anti-FLAG M1 (mouse monoclonal) | Sigma-Aldrich | #S3040 | (1:1000) |
| Antibody | Anti-TGN-38 (rabbit polyclonal) | Sigma-Aldrich | #T9826 | (1:1000) |
| Chemical compound, drug | SNAP-Cell 647 SiR | New England BioLabs | #S9102S | 1 µM; membrane -permeable SNAP tag substrate |
| Chemical compound, drug | SNAP-Surface 649 | New England BioLabs | #S9159S | 500 nM; membrane -impermeable SNAP tag substrate |
| Chemical compound, drug | Nerve growth factor (NGF) | Gibco | #13257 | 100 ng/ml; induces retention of newly synthesized DOR in the Golgi |
| Chemical compound, drug | LY294002 | Tocris | #1130 | 10 µM; PI3K inhibitor; induces retention of newly synthesized DOR in the Golgi |
| Chemical compound, drug | MI 14 | Tocris | #5604 | 20 µM |
| Chemical compound, drug | SNC80 | Tocris | #0764 | Small molecular DOR agonist |
| Chemical compound, drug | DPDPE | Tocris | #1431 | Peptide DOR agonist |
| Chemical compound, drug | ICI174864 | Tocris | #0820 | 100 µM; Peptide DOR inverse agonist |
| Chemical compound, drug | β-chlornaltrexamine (CNA) | Sigma-Aldrich | #O001 | 10 µM; irreversible antagonist |
| Chemical compound, drug | Cycloheximide (CHX) | Tocris | #0970 | 3 µg/ml; protein synthesis inhibitor |
| Chemical compound, drug | Forskolin | Sigma-Aldrich | #F3917 | 2 µM; adenylyl cyclase activator |

## DNA constructs

SSF-DOR construct consists of an N-terminal signal sequence followed by a Flag tag followed by the mouse DOR sequence in a pcDNA3.1 vector backbone. To create SNAP-DOR, the full-length receptor sequence was amplified from the SSF-DOR construct by PCR with compatible cut sites (BamHI and XbaI). The SNAP tag (New England Biolabs, Ipswich, MA) was amplified by PCR with compatible cut sites (HindIII and BamHI) and both were ligated into a pcDNA3.1 vector backbone to produce the final construct containing an N-terminal signal sequence, followed by the SNAP tag and then the receptor. β-arrestin-1 was generated from a geneblock (Integrated DNA Technologies, Coralville, IA) containing the human cDNA (ENST00000420843) for hARRB1 with HindIII and AgeI cut sites. mScarlet was amplified by PCR from pmScarlet_alphaTubulin_C1, a gift from Dorus Gadella (Addgene plasmid #85045) (*Bindels et al., 2017*), with AgeI and XbaI cut sites. Both were then ligated into a pcDNA3.1 vector backbone to produce a C-terminally tagged β-arrestin-1. β-arrestin-2 tagged with tdTomato was generated from β-arrestin 2-GFP via restriction site cloning (*Weinberg et al., 2017*). Nb39-mVenus was a gift from Drs. Bryan Roth and Tao Che (*Che et al., 2020*). Venus-miniGsi and

Venus-miniGs were gifts from Drs. Greg Tall and Nevin Lambert. pcDNA3-ICUE3 was a gift from Dr. Jin Zhang (Addgene plasmid #61622) (*DiPilato and Zhang, 2009*).

## Cell culture and transfection

PC12 cells (ATCC, #CRL-1721), which were validated cells purchased from ATCC, were used for all experiments. Cells in the lab were routinely tested for mycoplasma contamination, and only negative cells were used. Cells were maintained at 37°C with 5% $CO_2$ and cultured in F-12K media (Gibco, #21127), with 10% horse serum and 5% fetal bovine serum (FBS). Cells were grown in flasks coated with CollagenIV (Sigma-Aldrich, #C5533) to allow for adherence. PC12 cells were transiently transfected at 90% confluency according to manufacturer's guidelines with Lipofectamine 2000 (Invitrogen, #11668) with 1.5 ug of each DNA construct to be expressed. The transfection mixture was incubated with cells in Opti-MEM media (Gibco, #31985) for 5 hr, then removed and replaced with normal culture media until imaging 48–72 hr following transfection.

## Live-cell imaging with fluorescent biosensors

PC12 cells transfected with SNAP-DOR and the appropriate biosensor were plated and imaged in single-use MatTek dishes (MatTek Life Sciences, #P35G-1.5–14 C) coated with 20 µg/ml poly-D-lysine (Sigma-Aldrich, #P7280) for 1 hr. For experiments requiring a Golgi pool of DOR, cells were pretreated with 100 ng/ml of NGF (Gibco, #13257) or 10 µM LY294002 (Tocris, #1130) or 20 µM PI4K inhibitor MI 14 (Tocris, #5604) for 1 hr prior to imaging, as described previously (*Kim and von Zastrow, 2003*; *Shiwarski et al., 2017a*; *Shiwarski et al., 2017b*). Cells were labeled with 500 nM SNAP-Surface 649 (New England Biolabs, #S9159S) for 5 min at 37°C for TIR-FM imaging or 1 µM permeable SNAP-Cell 647-SiR (New England Biolabs, #S9102S) for 15 min followed by a 15-min wash in cell culture media for confocal imaging. Cells were imaged on a Nikon TiE inverted microscope using a x60/1.49 Apo-TIRF (Nikon Instruments, Melville, NY) objective in $CO_2$-independent Leibovitz's L-15 media (Gibco, #11415), supplemented with 1% FBS in a 37°C-heated imaging chamber (In Vivo Scientific). Red fluorescent protein (RFP) (β-arrestin-1 and β-arrestin-2; 561 nm excitation, 620 emission filter), yellow fluorescent protein (YFP) (Nb39-mVenus and Venus-miniGsi; 488 nm excitation, 446/523/600/677 quad-band filter), and the SNAP-labeled DOR (647 nm excitation, 700 emission filter when imaged with RFP or 446/523/600/677 quad-band filter when imaged with YFP) were excited with solid-state lasers and collected with an iXon +897 EMCCD camera (Andor, Belfast, UK).

## Immunofluorescence and fixed-cell imaging

PC12 cells transfected with Flag-DOR and either Nb39-mVenus or Venus-miniGsi were plated on poly-D-lysine-coated coverslips and grown at 37°C for 48 hr. To induce intracellular accumulation of newly synthesized DOR, cells were treated with NGF (100 ng/ml) for 1 hr prior to treatment for 15 min with 10 µM β-chlornaltrexamine (CNA; Sigma-Aldrich, #O001) or CNA followed by 10 µM SNC80 (Tocris, #0764) for 5 min. Cells were fixed with 4% paraformaldehyde, pH 7.4, for 20 min at 25°C followed by blocking with phosphate-buffered saline with 5% FBS, 5% glycine, 0.75% Triton-X-100, 1 mM magnesium chloride, and 1 mM calcium chloride. Primary and secondary antibody incubations were performed for 1 hr at 25°C in blocking buffer with anti-Flag-M1 (Sigma-Aldrich, #F3040, 1:1000) conjugated with Alexa-647 (Molecular Probes, #A20186) and anti-TGN-38 rabbit polyclonal antibody (Sigma-Aldrich, #T9826, 1:1000), and goat anti-Rabbit IgG conjugated to Alexa-568 (ThermoFisher, #A-11011, 1:1000), respectively. Cells were washed with blocking buffer without Triton-X-100 after primary and secondary incubations. Coverslips were mounted on glass slides using Prolong Diamond Reagent (Molecular Probes, #P36962). Cells were imaged on a Nikon TiE inverted microscope using a x60/1.49 Apo-TIRF (Nikon Instruments, Melville, NY) objective and iXon +897 EMCCD camera (Andor, Belfast, UK).

## Live-cell FRET imaging

PC12 cells transfected with SNAP-DOR and the ICUE3 FRET sensor were plated and imaged in MatTek dishes (MatTek Life Sciences, #P35G-1.5–14 C) coated with poly-D-lysine. For experiments comparing inhibition of Fsk-stimulated cAMP in cells with and without Golgi DOR, cells in all conditions were treated with CHX (3 µg/ml) for 1 hr before imaging to chase out any receptor transiting through the biosynthetic pathway. To induce a Golgi pool, cells were treated with NGF for 1 hr prior

to CHX treatment to build up a pool of internal receptors that is maintained even in the absence of new protein synthesis with NGF maintained in the media during the subsequent CHX incubation (*Shiwarski et al., 2017a*). CHX, NGF, and 100 µM ICI174864 (ICI; Tocris, #0820), when appropriate, were present in the media for the duration of the experiment. For experiments comparing the signaling of DPDPE (Tocris, #1431) peptide agonist and SNC80 small molecule agonist with and without a surface block, cells in all conditions were treated with 10 µM LY294002 for 1 hr prior to imaging to induce a Golgi pool of receptor, and LY294002 was maintained in the media throughout the duration of the experiment. Cells were labeled with 1 µM permeable SNAP-Cell 647-SiR for 15 min followed by a 15-min wash to visualize receptor. Cells were imaged in L-15 media supplemented with 1% FBS at 25℃ in a temperature-controlled imaging chamber (In Vivo Scientific) at 60 s intervals. Imaging was conducted on a Ti2 inverted microscope (Nikon Instruments, Melville, NY) with a x60 NA 1.49 Apo-TIRF objective (Nikon Instruments, Melville, NY). CFP (405 nm excitation, 400 emission filter), YFP (FRET) (405 nm excitation, 514 emission filter) and the SNAP-tagged isoform (647 nm excitation, 700 emission filter) were collected with an iXon-888 Life EMCCD camera (Andor, Belfast, UK) every 60 s with five frames of baseline after agonist addition before 2 µM Fsk (Sigma-Aldrich, #F3917) addition to stimulate adenylyl cyclase activity.

## Image quantification

All image quantifications were performed using ImageJ/Fiji (National Institutes of Health, Bethesda, MD) (*Rueden et al., 2017*; *Schindelin et al., 2012*). To quantify biosensor recruitment to IC DOR or PM DOR, the receptor channel at each timepoint was thresholded and used to create a binary mask to isolate only pixels containing the receptor signal. The receptor mask from each timepoint was then applied to the corresponding timepoint in the biosensor channel to produce an image of biosensor fluorescence in regions of the cell containing the receptor. A region of interest corresponding to intracellular receptor was selected in confocal images, and in TIRF images, a region of interest capturing the entire cell was selected. Mean fluorescence intensity was then measured in these images over time and normalized to average baseline fluorescence before drug addition.

A similar approach was used to measure biosensor recruitment to the Golgi in cells fixed and stained for TGN-38. In these images, TGN-38 was used to create the binary mask, which was then applied to the biosensor channel to isolate biosensor fluorescence in the Golgi region of the cell. An inverse mask of the TGN-38 channel was also created and applied to the biosensor channel to isolate biosensor fluorescence in all other regions of the cell. Biosensor enrichment in the Golgi is expressed as the mean fluorescence intensity in the Golgi region divided by the mean fluorescence intensity in the rest of the cell. FRET images were analyzed in ImageJ as previously described (*Shiwarski et al., 2017a*; *Weinberg et al., 2017*). Briefly, the CFP channel was divided by the FRET channel at each timepoint. A region of interest was defined for each cell in a given field and the resulting CFP/FRET ratio measured at each timepoint. The CFP/FRET ratio was normalized to the mean baseline ratio before drug addition for each cell. Endpoint CFP/FRET ratios (measured as the change of the endpoint value from the baseline of 1) and total cAMP responses (measured as area under the curve) for all cells were normalized to the average of cells in the control condition for each experimental replicate.

## Acknowledgements

We would like to thank Dr. Alan Smrcka, Dr. Lois Weisman, Dr. Bing Ye, and Candilianne Serrano Zayas for their valuable feedback on this project. We also thank Drs. Bryan Roth, Tao Che, Greg Tall, and Nevin Lambert for essential reagents. Funding: SEC was supported by National Science Foundation Graduate Research Fellowship under Grant DGE 1256260. MAP was supported by NIH Grant GM117425 and National Science Foundation (NSF) grant 1935926.

## Additional information

### Funding

| Funder | Grant reference number | Author |
|---|---|---|
| National Science Foundation | DGE 1256260 | Stephanie E Crilly |
| National Institute of General Medical Sciences | GM117425 | Manojkumar A Puthenveedu |
| National Science Foundation | 1935926 | Manojkumar A Puthenveedu |

The funders had no role in study design, data collection and interpretation, or the decision to submit the work for publication.

### Author contributions

Stephanie E Crilly, Conceptualization, Formal analysis, Validation, Investigation, Visualization, Methodology, Writing - original draft, Writing - review and editing; Wooree Ko, Resources, Formal analysis, Investigation, Methodology, Writing - review and editing; Zara Y Weinberg, Resources, Data curation, Software, Formal analysis, Writing - review and editing; Manojkumar A Puthenveedu, Conceptualization, Resources, Formal analysis, Supervision, Funding acquisition, Visualization, Methodology, Project administration, Writing - review and editing

### Author ORCIDs

Stephanie E Crilly (iD) https://orcid.org/0000-0002-8151-290X
Zara Y Weinberg (iD) https://orcid.org/0000-0001-7176-038X
Manojkumar A Puthenveedu (iD) https://orcid.org/0000-0002-3177-4231

### Decision letter and Author response

Decision letter https://doi.org/10.7554/eLife.67478.sa1
Author response https://doi.org/10.7554/eLife.67478.sa2

## Additional files

### Supplementary files

- Source data 1. Numerical data file.
- Transparent reporting form

### Data availability

All data generated and analyzed have been included in this study. No new sequencing data or structural data are reported.

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
