## [Decision Letter]

**Acceptance summary:**

This paper is of interest for cell biologists and pharmacologists interested in G protein-coupled receptor signaling. There is a growing body of literature showing that drug receptors signal differently from the plasma membrane and intracellular compartments. This work provides new insight into how the subcellular location of one such receptor, the δ opioid receptor, can lead to differential sensitivity to external stimuli. The approach is technically sophisticated, making use of novel biosensors for different receptor conformations. Overall, the data is intriguing, furthering our understanding of how location can "bias" GPCR signaling. The results set the stage for more mechanistic studies that might seek to determine how different cellular environments affect responses to drugs.

**Decision letter after peer review:**

Thank you for submitting your article "Conformational specificity of opioid receptors is determined by subcellular location irrespective of agonist" for consideration by *eLife*. Your article has been reviewed by 3 peer reviewers, and the evaluation has been overseen by Jonathan Cooper as the Senior and Reviewing Editor. The following individuals involved in review of your submission have agreed to reveal their identity: Aashish Manglik (Reviewer #1); Nevin A Lambert (Reviewer #2); John T Williams (Reviewer #3).

Essential revisions:

The review discussion mostly focused on the convergence of the independent reviews on the same conclusion: that this paper is an importance advance for the field. The individual "Recommendations for the Authors" are below, and you will see that they can be simply addressed by:

1. Adding quantification of recruitment of probes to the plasma membrane vs Golgi.

2. Concentration-response curves, to allow for the possibility that drug concentrations at Golgi may be much lower than at the PM.

3. Add a caveat to the Discussion regarding the possibility that differential drug concentrations at PM and Golgi may contribute to the conformational specificity of the receptors.

*Reviewer #1:*

Crilly et al. investigated how the cellular location of a G protein-coupled receptor (GPCR) influences its ability to respond to an activating drug. As a model system, they use the δ-opioid receptor (DOR), a well studied GPCR with potential roles in pain biology. To directly assess where the DOR is active in cells, they make use of two previously developed biosensors that specifically bind activated opioid receptors. One of these biosensors is a nanobody (Nb39) that was used to solve the structure of active mu-opioid receptor. Another biosensor is an engineered "mini" G protein that binds activated Gi-coupled receptors. Intriguingly, they find that although both of these biosensors bind to activated DOR when it is located at the plasma membrane, only Nb39 binds to DOR when it is located at the Golgi membrane. By using a combination of membrane impermeant and permeant DOR ligands, they specifically isolate that this effect is unique to the Golgi pool and independent of whether plasma membrane DOR is activated. The GPCR effector β-arrestin is conversely only specific to the plasma membrane DOR. Despite this distinction in which proteins can be recruited to DOR depending on sub cellular location, both plasma membrane and Golgi DOR can interact with G proteins to inhibit cAMP production.

The central conclusion of this work is that the differential localization of these two biosensors in response to the same agonist (SNC80) suggests that the conformation of DOR is different when its at the plasma membrane vs the Golgi. The experiments carried out in this work are of high quality, and the individual conclusions are supported by the data. However, the overall conclusion of this work relies on the notion that binding of the minGsi biosensor to DOR is physiologically meaningful. Clearly both Nb39 and wild-type Gi (presumably leading to the effect on cAMP) are able to bind DOR in both locations. Arrestin recruitment is known to be specific to plasma membrane, which is defined in part by activity of the GRK2 kinase at this specific location. That leaves only the miniGsi "transducer" as the outlier, with its incapability to recruit to the Golgi membrane. While the data are indeed intriguing, it is possible that this distinction may just be specific to miniGsi. The authors should entertain the possibility that this differential recruitment of minGsi may result from other factors apart from a distinct DOR conformation. For example, it is possible that different intracellular concentrations of sensors in cells and their relative affinities for a similar active state may yield similar results. This may be especially true in PC12 cells, where the confocal signal for plasma membrane recruitment of both Nb39 and miniGsi is fairly weak. Better quantitation of relative recruitment of each biosensor to the plasma membrane and Golgi in the same cell line (e.g. PC12) would help some of these concerns.

The imaging work is fantastic, but I think more caveats are needed in the discussion to highlight that the differential recruitment of miniGsi may happen for other reasons. It would be helpful if there was a way to quantitate the level of plasma membrane activation by miniGsi and Nb39 in PC12 cells, as the confocal images in 1H/I suggest that there is barely any recruitment to this compartment.

*Reviewer #2:*

Crilly and colleagues study recruitment of two different G protein surrogate "conformational sensors" to active DORs at the plasma membrane and Golgi apparatus. This is similar to several recent studies examining the activity of GPCRs at different subcellular locations, and is particularly interesting in this case because a substantial population of DORs resided in the Golgi in neurons. They make the surprising observation that a nanobody sensor (Nb39) is recruited to DORs at both the PM and Golgi in response to a membrane-permeant agonist, whereas a mini G protein sensor (mini Gsi) is recruited only to the PM in response to the same ligand. They further show that this ligand fails to recruit arrestin to the Golgi, but does appear to activate Gi heterotrimers in this compartment, as assessed by inhibition of adenylyl cyclase in the presence of an impermeant antagonist. They conclude that subcellular location affects conformational sampling of the receptor, although the underlying mechanism is unknown.

The major strengths of the manuscript include the striking and surprising main result, namely a dramatic difference in the response of two biosensors that might be expected to behave identically. Moreover, the authors extend the observation to show apparent activation of Gi heterotrimers in the Golgi, which makes the primary observation even more surprising and is a nice result in its own right. Finally, the manuscript is succinct and beautifully written, and the experiments are carefully controlled and logically laid out.

Weaknesses of the manuscript include the absence of a molecular mechanism, although admittedly this is a tall order as there are many possibilities. Mitigating this somewhat, the discussion contains appropriate speculation about what factors in "the immediate subcellular environment" might affect conformational sampling. It must also be said that conformational sampling is not observed directly, but is inferred from the behavior of two different probes that could conceivably have other properties that differentially affect movement between compartments. There are also a few technical issues, and a few interesting observations that deserve more discussion.

These results are a valuable addition to the growing body of literature addressing the possible functional roles of GPCRs located in intracellular compartments, and set the stage for more mechanistic studies that might seek to determine how different lipid environments (for example) in cells affect GPCR conformational dynamics, or motivate studies seeking to monitor receptor conformation more directly in intact cells.

The authors should produce a concentration-response curve for both sensors at the PM. If SNC80 is much less potent for mini Gsi recruitment then it is possible that the difference observed at the Golgi is due to inadequate SNC80 exposure to the Golgi lumen. This is important, as the concentration of SNC80 in the Golgi lumen is unknown.

Can the authors produce a DOR/sensor fluorescence ratio in different ROIs centered on PM and Golgi? Recruitment efficiency of both sensors to the PM seems to be very low for SNC80 in general. The confocal images in Figure 1 show very modest recruitment of either sensor to a significant pool of PM DOR (c.f. arrestin), and the only really compelling images of either sensor at the PM are TIRF.

SNC80 produces only a slow 50% increase in Nb39 at the PM, whereas the same ligand produces a faster 3-fold increase in mini Gsi at the PM. Perhaps it's worth pointing out that not only does the "mini Gsi conformation" of the receptor seem to be absent at the Golgi, but it also seems to be preferred at the PM. This is perhaps subtle but not clear from the present discussion.

Could the authors quantify the relative surface vs. Golgi pools of DOR under the conditions used in their experiments and report this? As it is the manuscript sort of implies that cells have DOR either almost entirely at the PM (without NGF, etc.) or almost entirely at the Golgi (with NGF), and this seems unlikely. This might have important implications. For example, on page 4, line 23 the authors attribute a DPDPE vs. SNC80 difference to ligand bias, but alternatively DPDPE recruitment of Nb39 to the PM may be stronger due to lack of competition from receptors in intracellular sites, which presumably are present at some level after overexpression despite the lack of NGF pretreatment.

Cycloheximide was used for the cAMP experiments. This is only in the methods, and should be mentioned and justified (why was this necessary?) in the main text.

Has any receptor been shown to recruit mini Gsi to the Golgi? If a positive control for this can be shown or cited it would make a more useful control than mini Gs.

It is a bit odd that 10 μm SNC80 was used for the only experiments showing recruitment to the PM, and 100 fold less was used thereafter.

2 μm Fsk is used. This seems like a modest or weak AC stimulus. Was this critical for observing DOR effects?

*Reviewer #3:*

This manuscript uses state of the art tools in a clever series of experiments to investigate the roll of intracellular receptors. It has been known for some time that cell permeant ligands associate with receptors located in the Golgi. This work demonstrates that the active conformation or at least or one of the active conformations of DOR is distinctly different in the two locations. This is an important contribution.

The demonstration that the Golgi associated receptors are functional is important. Again this is known from work with Gs linked receptors but remains an important observation. This observation suggests that although the miniGsi does not mimic the conformation that some G protein does.

Finally the discussion on the fact that active states of the receptor are highly dynamic is very important and often not appreciated.

In the first read though the manuscript I found the presentation of the functional part a little confusing. With a more careful read it was clear that the inhibition of adenylyl cyclase induced by intracellular receptors was very significant. Does that mean that the plasma membrane associated receptors do not inhibit cyclase? Seems to be something that should be addressed.

The first paragraph of the discussion blew me away with the details. It is clearly important to distinguish how the sensors associate with the receptor. I think a more gentle description of why that is important to begin with would help readers that are not intimately knowledgeable of GPCR structure.

---

## [Author Response]

Essential revisions:The review discussion mostly focused on the convergence of the independent reviews on the same conclusion: that this paper is an importance advance for the field. The individual "Recommendations for the Authors" are below, and you will see that they can be simply addressed by:1. Adding quantification of recruitment of probes to the plasma membrane vs Golgi.

We have added quantification of both probes to the plasma membrane (Figure 1M) in confocal images in the same cells from which we quantified sensor recruitment to the Golgi (Figure 1L). When quantitated, we see an increase in miniGsi and Nb39 recruitment to PM DOR, although the Nb39 increase was smaller. The plasma membrane recruitment is less obvious in confocal compared to TIR-FM imaging, as expected based on our experience imaging with these sensors, but the increases match the trends we see in TIR-FM (Figure 1A-F). This shows that in the same cells in which we see no miniGsi recruitment to IC DOR, we still see recruitment to PM DOR.

2. Concentration-response curves, to allow for the possibility that drug concentrations at Golgi may be much lower than at the PM.

We appreciate the concern that, if miniGsi were recruited less efficiently than Nb39, lower drug concentrations at the Golgi could result in an apparent lack of miniGsi reduction. To address this, we have added concentration response curves for Nb39 and miniGsi recruitment to PM DOR (Figure 1G) in response to increasing concentrations of SNC80. The concentration response curve for miniGsi is leftshifted compared to Nb39, suggesting that SNC80 is more potent at recruiting miniGsi. Therefore, we think it is unlikely that the differential recruitment of miniGsi and Nb39 to IC DOR is a result of lower drug concentrations in the intracellular compartment. We thank you for suggesting this experiment, and we have added this point to the revised manuscript.

3. Add a caveat to the Discussion regarding the possibility that differential drug concentrations at PM and Golgi may contribute to the conformational specificity of the receptors.

In the revised discussion, we have discussed the caveat that drug concentrations in the Golgi may be different from extracellular concentrations in the context of our results that miniGsi is recruited more efficiently than Nb39 to the plasma membrane.

In addition to these revisions, we have also clarified some of the technical questions and additional discussion points raised by the reviewers in the revised manuscript, as described below. We thank the reviewers for recommending these changes, which we believe have greatly strengthened our manuscript.

Reviewer #1:Crilly et al. investigated how the cellular location of a G protein-coupled receptor (GPCR) influences its ability to respond to an activating drug. As a model system, they use the δ-opioid receptor (DOR), a well studied GPCR with potential roles in pain biology. To directly assess where the DOR is active in cells, they make use of two previously developed biosensors that specifically bind activated opioid receptors. One of these biosensors is a nanobody (Nb39) that was used to solve the structure of active mu-opioid receptor. Another biosensor is an engineered "mini" G protein that binds activated Gi-coupled receptors. Intriguingly, they find that although both of these biosensors bind to activated DOR when it is located at the plasma membrane, only Nb39 binds to DOR when it is located at the Golgi membrane. By using a combination of membrane impermeant and permeant DOR ligands, they specifically isolate that this effect is unique to the Golgi pool and independent of whether plasma membrane DOR is activated. The GPCR effector β-arrestin is conversely only specific to the plasma membrane DOR. Despite this distinction in which proteins can be recruited to DOR depending on sub cellular location, both plasma membrane and Golgi DOR can interact with G proteins to inhibit cAMP production.The central conclusion of this work is that the differential localization of these two biosensors in response to the same agonist (SNC80) suggests that the conformation of DOR is different when its at the plasma membrane vs the Golgi. The experiments carried out in this work are of high quality, and the individual conclusions are supported by the data. However, the overall conclusion of this work relies on the notion that binding of the minGsi biosensor to DOR is physiologically meaningful. Clearly both Nb39 and wild-type Gi (presumably leading to the effect on cAMP) are able to bind DOR in both locations. Arrestin recruitment is known to be specific to plasma membrane, which is defined in part by activity of the GRK2 kinase at this specific location. That leaves only the miniGsi "transducer" as the outlier, with its incapability to recruit to the Golgi membrane. While the data are indeed intriguing, it is possible that this distinction may just be specific to miniGsi. The authors should entertain the possibility that this differential recruitment of minGsi may result from other factors apart from a distinct DOR conformation. For example, it is possible that different intracellular concentrations of sensors in cells and their relative affinities for a similar active state may yield similar results. This may be especially true in PC12 cells, where the confocal signal for plasma membrane recruitment of both Nb39 and miniGsi is fairly weak. Better quantitation of relative recruitment of each biosensor to the plasma membrane and Golgi in the same cell line (e.g. PC12) would help some of these concerns.

We are very happy that the reviewer felt the experiments were of high quality and that they supported the conclusions of the work. We also thank them for raising possible explanations for these findings including sensor concentrations and relative affinities that were not explicitly discussed in the original manuscript. We have now included new data and rewritten the discussion to address these comments.

To address the possibility that different intracellular concentrations of sensors and affinities in cells may explain the differential recruitment we have added three new pieces of data to the revised manuscript.

First, as the reviewers requested, we quantified recruitment of both sensors to PM DOR by confocal imaging (Figure 1M) in the same cells in which we quantified recruitment to IC DOR. We see miniGsi recruitment to the PM in the same cells where we do not see recruitment to Golgi.

Second, we plotted the maximum sensor recruitment to IC DOR against sensor expression of each sensor to look for a linear relationship between the sensor expression and recruitment (Figure 1 supplemental figure 3 C-D). The maximum recruitment to IC DOR was not correlated with sensor expression for either sensor. For miniGsi in particular, because it showed differential recruitment, we imaged cells across a broad range of expression levels. We saw no miniGsi recruitment to IC DOR at any expression level across this range.

Third, we generated concentration-response curves for SNC80 sensor recruitment, as described above. At the plasma membrane, the curve for miniGsi is left-shifted relative to the curve for Nb39. We could not estimate the dose response of miniGsi on the Golgi because we see practically no recruitment. The lack of miniGsi recruitment to the Golgi is the opposite of what we would expect if the differential recruitment were driven solely by concentrations and affinities.

Together, therefore, we think that the differential recruitment we see is a function of the location. Of course, this could be driven by affinity of the sensors relative to different PM-localized or Golgi-localized endogenous interacting proteins at different locations.

The imaging work is fantastic, but I think more caveats are needed in the discussion to highlight that the differential recruitment of miniGsi may happen for other reasons. It would be helpful if there was a way to quantitate the level of plasma membrane activation by miniGsi and Nb39 in PC12 cells, as the confocal images in 1H/I suggest that there is barely any recruitment to this compartment.

We thank the reviewer for making this suggestion. As requested by the reviewers we have quantitated recruitment of both sensors to the plasma membrane in confocal images in the same cells from which we quantified sensor recruitment to intracellular compartments (Figure 1M). We see a significant increase in miniGsi recruitment to PM DOR and a smaller, though not significant, increase in Nb39 recruitment, matching the trends we see in TIRF imaging. This shows that in the same cells in which we see no miniGsi recruitment to IC DOR, we still see miniGsi recruitment to PM DOR, emphasizing the location specific recruitment of these biosensors. The TIR-FM imaging, which we have used to determine a dose response of PM recruitment of sensors (as was used by Stoeber et al), was a more sensitive method study recruitment of both sensors to the PM.

The relatively modest recruitment of these sensors to the PM, compared to intracellular compartments, by confocal imaging is consistent with previous work (Irannejad et al.,2013; Stoeber et al., 2018) and our experience imaging these sensors with a number of GPCRs. This could be due to rapid phosphorylation of and arrestin recruitment by the receptors at the PM. At present, we do not know the rate and extent of DOR phosphorylation on other compartments, but we show that there is minimal arrestin recruitment on the Golgi.

Reviewer #2:[…] The major strengths of the manuscript include the striking and surprising main result, namely a dramatic difference in the response of two biosensors that might be expected to behave identically. Moreover, the authors extend the observation to show apparent activation of Gi heterotrimers in the Golgi, which makes the primary observation even more surprising and is a nice result in its own right. Finally, the manuscript is succinct and beautifully written, and the experiments are carefully controlled and logically laid out.Weaknesses of the manuscript include the absence of a molecular mechanism, although admittedly this is a tall order as there are many possibilities. Mitigating this somewhat, the discussion contains appropriate speculation about what factors in "the immediate subcellular environment" might affect conformational sampling. It must also be said that conformational sampling is not observed directly, but is inferred from the behavior of two different probes that could conceivably have other properties that differentially affect movement between compartments. There are also a few technical issues, and a few interesting observations that deserve more discussion.These results are a valuable addition to the growing body of literature addressing the possible functional roles of GPCRs located in intracellular compartments, and set the stage for more mechanistic studies that might seek to determine how different lipid environments (for example) in cells affect GPCR conformational dynamics, or motivate studies seeking to monitor receptor conformation more directly in intact cells.

We are very happy that the reviewer thought the experiments were carefully controlled and logically presented, and found our results a valuable addition to the field. We agree that the mechanism behind the differential recruitment is, as of now, not known. It is a challenging but compelling direction for future research. We thank this reviewer and the other reviewers for offering additional potential mechanisms for the observed differential recruitment which we have discussed in more detail in the revised manuscript. We have also clarified the experimental details requested by the reviewer and have elaborated on the differential sensor recruitment we observe to PM DOR, which builds on similar recent work, in the revised manuscript.

The authors should produce a concentration-response curve for both sensors at the PM. If SNC80 is much less potent for mini Gsi recruitment then it is possible that the difference observed at the Golgi is due to inadequate SNC80 exposure to the Golgi lumen. This is important, as the concentration of SNC80 in the Golgi lumen is unknown.

We thank the reviewer for suggesting this experiment to test whether the lack of miniGsi recruitment to the Golgi is because SNC80 is less potent in general at recruiting miniGsi. We have added concentration response curves for Nb39 and miniGsi recruitment to PM DOR (Figure 1G) in response to increasing concentrations of SNC80. These curves indicate that miniGsi is recruited to PM DOR more potently than Nb39. These data roughly matched the relative trends with which miniGsi and a similar nanobody Nb33 were recruited to two related opioid receptors, kappa and mu opioid receptors, on the plasma membrane (Stoeber, et al. 2020). Because miniGsi is more potently recruited we think it is unlikely that the differential recruitment of miniGsi and Nb39 to IC DOR is a result of lower drug concentrations in the intracellular compartment. The reviewer is correct that the concentration of SNC80 in the Golgi is unknown. In the revised manuscript, we have discussed this possibility in the context of our new results.

Can the authors produce a DOR/sensor fluorescence ratio in different ROIs centered on PM and Golgi? Recruitment efficiency of both sensors to the PM seems to be very low for SNC80 in general. The confocal images in Figure 1 show very modest recruitment of either sensor to a significant pool of PM DOR (c.f. arrestin), and the only really compelling images of either sensor at the PM are TIRF.

We have quantitated (Figure 1M) the recruitment of both sensors to the plasma membrane in confocal images in the same cells from which we quantitated sensor recruitment to intracellular compartments. We see an increase in miniGsi recruitment to PM DOR and a smaller increase in Nb39 recruitment, matching the trends we see in TIRF imaging. Importantly, this shows that in the same cells in which we see no miniGsi recruitment to IC DOR, we still see significant recruitment to PM DOR, emphasizing that the recruitment of these biosensors is location specific. We have discussed these points in the revised manuscript.

SNC80 produces only a slow 50% increase in Nb39 at the PM, whereas the same ligand produces a faster 3-fold increase in mini Gsi at the PM. Perhaps it's worth pointing out that not only does the "mini Gsi conformation" of the receptor seem to be absent at the Golgi, but it also seems to be preferred at the PM. This is perhaps subtle but not clear from the present discussion.

We agree that this is an interesting and important result to point out. This comparison emphasizes the differential recruitment of miniGsi to DOR in different compartments. We have added a discussion of this point to our revised manuscript.

Could the authors quantify the relative surface vs. Golgi pools of DOR under the conditions used in their experiments and report this? As it is the manuscript sort of implies that cells have DOR either almost entirely at the PM (without NGF, etc.) or almost entirely at the Golgi (with NGF), and this seems unlikely. This might have important implications. For example, on page 4, line 23 the authors attribute a DPDPE vs. SNC80 difference to ligand bias, but alternatively DPDPE recruitment of Nb39 to the PM may be stronger due to lack of competition from receptors in intracellular sites, which presumably are present at some level after overexpression despite the lack of NGF pretreatment.

We thank the reviewer for raising these points. In PC12 cells, in the timeframe that we use for acute accumulation of an internal pool of DOR, the receptors initially at the surface predominantly remain at the surface. The cell therefore contains a substantial pool of PM DOR. We have clarified the description of relative pools of DOR in the revised manuscript.

It is possible that competition from intracellular DOR could complicate plasma membrane Nb39 recruitment for SNC80 but not for DPDPE. However, we think it is unlikely for several reasons. First, we use saturating doses of both drugs in our assays, so drug availability should not be limiting. Second, our data in Figure 1 supplemental figure 3B show similar levels of sensor recruitment to the PM in untreated cells compared to cells treated with NGF, in which we expect a much larger pool of internal receptors. Third, there is no correlation between sensor expression and relative recruitment to the Golgi (Figure 1 supplemental figure 3 C-D), suggesting that sensor is not limiting.

Cycloheximide was used for the cAMP experiments. This is only in the methods, and should be mentioned and justified (why was this necessary?) in the main text.

We thank the reviewer for pointing out this important detail. We treated cells with cycloheximide to both ensure that all DOR present within the biosynthetic pathway was cleared out before the experiment, which is especially relevant to conditions requiring PM DOR only, and also to prevent localization of newly synthesized DOR along the biosynthetic pathway during the course of the experiment. These experiments were conducted at 25°C which can slow biosynthetic transport. We felt it was important to prevent new synthesis of DOR which could accumulate in the Golgi under these conditions and potentially complicate our interpretation of conditions requiring no IC DOR. For the imaging experiments, because these were single-cell experiments with higher dynamic range, we felt that the cycloheximide was not critical. We have explained our use of cycloheximide for these experiments in the Results section of the revised manuscript.

Has any receptor been shown to recruit mini Gsi to the Golgi? If a positive control for this can be shown or cited it would make a more useful control than mini Gs.

We thank the reviewer for drawing our attention to this point. Wan et al., 2018, have shown, using BRET and imaging , that the Gi-coupled A1-adenosine receptor, when localized to the Golgi by overexpression, can recruit miniGsi to this compartment in response to adenosine treatment. We agree that this result is a great reference to show that miniGsi can in fact be recruited to Gi-coupled receptors in intracellular compartments for some receptors, but not others. We had cited this work, but not in this context. We have now discussed these results in the revised manuscript.

It is a bit odd that 10 μm SNC80 was used for the only experiments showing recruitment to the PM, and 100 fold less was used thereafter.

We used 10μM concentration for all drugs to ensure that we would be at saturating concentrations to see the maximum recruitment possible for all conditions. 10μM as a saturating concentration is supported by the concentration-response curves we have included in Figure 1G. We decreased the concentration of agonist for all experiments which required the peptide antagonist ICI because we wanted to ensure that ICI was present in excess relative to the agonist so that it would effectively block PM receptor sites.

2 μm Fsk is used. This seems like a modest or weak AC stimulus. Was this critical for observing DOR effects?

We used 2μM Fsk because it gave us the highest dynamic range for the assay. We have previously tested different doses of Fsk for the ICUE3 sensor in PC12 cells, and 2µM Fsk gave us a 15-20% increase in CFP/FRET (which was the same response we got at 5µM, and we believe is close to the maximum change for the sensor). We saw 70-80% inhibition of this response with 100nM SNC80, which provided us with a sufficient dynamic range to observe intermediate effects. 2µM also matches the dose previously used in similar experiments (Stoeber et al., 2018).

Reviewer #3:[…] In the first read though the manuscript I found the presentation of the functional part a little confusing. With a more careful read it was clear that the inhibition of adenylyl cyclase induced by intracellular receptors was very significant. Does that mean that the plasma membrane associated receptors do not inhibit cyclase? Seems to be something that should be addressed.

We thank the reviewer for raising this point, and we think it is important to make sure that this point is clarified. We believe that both PM DOR and IC DOR inhibit cAMP. When compared, IC DOR suppresses cAMP to approximately half the degree suppressed by combined PM and IC DOR. We have revised the manuscript to present these data as clearly as possible. We have also provided source data that shows the percentage numbers presented in the graph.

In clarifying the signaling data, we also made a minor change to the presentation of endpoint Fsk responses in Figure 4J and Figure S4H. We had previously represented the endpoint Fsk responses as the normalized endpoint ratio (eg. 1.15), whereas we have now represented the endpoint Fsk responses as the change or δ in the normalized endpoint ratio from the baseline of 1 (eg. 0.15). This representation does not change the overall trends or comparative degree of suppression between conditions, but more accurately depicts the degree of suppression of the endpoint Fsk responses in each condition.

The first paragraph of the discussion blew me away with the details. It is clearly important to distinguish how the sensors associate with the receptor. I think a more gentle description of why that is important to begin with would help readers that are not intimately knowledgeable of GPCR structure.

We thank the reviewer for this suggestion. We have modified the beginning of the discussion to describe that location-biased signaling is an important new concept, and that biosensors are critical to study this concept.